# The Cytotoxic Effect of Isolated Cannabinoid Extracts on Polypoid Colorectal Tissue

**DOI:** 10.3390/ijms231911366

**Published:** 2022-09-26

**Authors:** Dana Ben-Ami Shor, Ilan Hochman, Nathan Gluck, Oren Shibolet, Erez Scapa

**Affiliations:** 1Department of Gastroenterology, Tel-Aviv Sourasky Medical Center, Tel-Aviv, Israel, Affiliated to Sackler Faculty of Medicine, Tel-Aviv University, Tel Aviv 6997801, Israel; 2CNBX Pharmaceuticals Ltd., Rehovot 7608801, Israel

**Keywords:** cannabis, cannabinoids, colorectal polyps, synergism, cytotoxicity

## Abstract

Purified cannabinoids have been shown to prevent proliferation and induce apoptosis in colorectal carcinoma cell lines. To assess the cytotoxic effect of cannabinoid extracts and purified cannabinoids on both colorectal polyps and normal colonic cells, as well as their synergistic interaction. Various blends were tested to identify the optimal synergistic effect. Methods: Biopsies from polyps and healthy colonic tissue were obtained from 22 patients undergoing colonic polypectomies. The toxicity of a variety of cannabinoid extracts and purified cannabinoids at different concentrations was evaluated. The synergistic effect of cannabinoids was calculated based on the cells’ survival. Isolated cannabinoids illustrated different toxic effects on the viability of cells derived from colorectal polyps. THC-d8 and THC-d9 were the most toxic and exhibited persistent toxicity in all the polyps tested. CBD was more toxic to polypoid cells in comparison to normal colonic cells at a concentration of 15 µM. The combinations of the cannabinoids CBDV, THCV, CBDVA, CBCA, and CBGA exhibited a synergistic inhibitory effect on the viability of cells derived from colon polyps of patients. Isolated cannabinoid compounds interacted synergistically against colonic polyps, and some also possessed a differential toxic effect on polyp and adjacent colonic tissue, suggesting possible future therapeutic value.

## 1. Introduction

Colorectal cancer (CRC) is the third most common cancer affecting both males and females in the United States [1] despite a steady decline in incidence and mortality rates, which can be attributed to colonoscopy screening and the preventive resection of adenomatous polyps [2]. CRC is a heterogeneous disease that differs in clinical presentation, molecular characteristics, and prognosis [3]. A series of molecular and histopathological changes leads the normal colonic epithelial cells to form aberrant crypt foci (ACF) and colorectal polyps that can further develop into CRC [4,5]. Endoscopic resection (i.e., polypectomy), the primary therapeutic intervention for colorectal polyps, can abort the processes of CRC development and reduce the prevalence of CRC [5].

Lesions ≥ 20 mm, whether sessile or flat and laterally spreading, are forms of advanced mucosal neoplasia and are considered high-risk precursors of CRC [6,7,8]. Surgery or the use of advanced polypectomy techniques such as endoscopic mucosal resection (EMR), endoscopic submucosal dissection (ESD), or hybrid techniques may be required in these cases [7]. Despite advances in resection techniques, recurrence following the EMR of advanced mucosal neoplasia ranges from 10% to 30% [9,10]. As the natural history of CRC is protracted, clinical trials have concentrated on the prevention of the recurrence of polyps at the polypectomy site in order to prevent their malignant transformation into adenocarcinoma [11]. Decreasing recurrence could be addressed through better resection techniques (e.g., ESD or resection margin ablation), and could be augmented through the creation of a microenvironment unfavorable to cancer growth. Chemoprevention has been shown to decrease polyp appearance and recurrence in various studies [12,13,14].

Cannabinoids are a group of compounds extracted from the cannabis plant. They include endocannabinoids, phytocannabinoids, and synthetic cannabinoids [15]. The pharmacological activity of cannabinoids is mediated by G-protein-coupled receptors (GPCRs): cannabinoid receptor 1 (CB1) and cannabinoid receptor 2 (CB2) [16]. An increasing number of studies have shown that phytocannabinoids can prevent proliferation, and induce apoptosis in a variety of cancer cell types, including breast, lung, prostate, skin, intestine, glioma, and others [17]. The ability to regulate signaling pathways that are critical for cell growth and survival is what makes these actions possible [17].

In a previous study, tetrahydrocannabinol (THC) treatment induced apoptosis in a CB1-dependent pathway in CRC cells by inhibiting various survival signaling cascades, while activating the proapoptotic BCL-2 family member BAD [18]. Additionally, cannabidiol (CBD) reduces cell proliferation in CRC cell lines. In an animal model, it reduced ACF (preneoplastic lesions), polyp, and tumor formation and counteracted colon cancer-induced changes in gene expression [19]. A CBD-rich cannabis extract was also shown to inhibit CRC cell proliferation and to attenuate colon carcinogenesis [20]. This activity involved both CB1- and CB2-receptor activation [20]. Cannabigerol (CBG) was found to promote apoptosis, stimulate reactive oxygen species (ROS) production, and reduce cell growth in CRC cells. In vivo, CBG inhibited the growth of chemically induced colon carcinogenesis and xenograft tumors [21]. It has been proposed that botanical extraction is more effective in achieving the desired therapeutic outcome than the synthetic ingredient, as the combination of compounds may have a synergistic effect [22].

Despite the accumulating literature regarding the anticancer action of cannabinoids, there is little data regarding their cytotoxic activity on colorectal polyps [23], and their potential therapeutic role in local chemoprevention of polyp recurrence at polypectomy sites or in the prevention of malignant transformation. In this study, we aimed to evaluate the cytotoxic effect of cannabinoid extracts on colorectal polyps versus normal colonic cells, as well as their synergistic interaction. Various blends were tested in order to try to find the optimal synergistic effect.

## 2. Results

P001-P022 designate the various patients’ polyps. Two patients were excluded from the study analysis due to borderline-positive hepatitis A virus (HAV) serology [patients’ polyps’ number 11 and 15 (P011 and P015)]. Cannabinoid extracts, as well as isolated cannabinoids, were tested on patients’ polyps and adjacent colonic mucosa numbered 1 and 2 (P001-P002). Isolated cannabinoids alone were tested on patients’ polyps and adjacent mucosa numbered 3–10 (P003–P010). P003 was used for calibration. Patients’ polyps numbered 12–22 (P012–P014 and P016–P022) were evaluated for synergistic interaction in inhibiting the viability of colon-polyp-derived cells by various isolated cannabinoids. Patient and polyp characteristics are presented in Table 1.

### 2.1. Evaluation of the Toxicity of Cannabinoids on Colon-Polyp-Derived Cells

The isolated cannabinoids’ effect on the viability of the polyp-derived cells was evaluated and is presented in Table 2. The isolated cannabinoids THC-d8 and THC-d9 (THC predominant isomeric forms) were the most toxic and exhibited persistent toxicity in all the polyps tested. CBCA and CBGA were the least toxic. Furthermore, some cannabinoids exhibited a broad variability in their capacity to elicit toxicity on different polyps.

### 2.2. Evaluation of the Efficacy of Cannabinoids on the Viability of Cells Derived from Patient Colon Polyps and Adjacent Normal Cells

Polyp and adjacent normal colonic tissues were obtained following polyp resection. The differential toxic effect of the various cannabinoids extracts as well as of the isolated cannabinoids was examined. Most of the cannabinoids showed similar toxicity to polyp or adjacent colonic tissue. However, cannabidiol (CBD) exerted a differential toxic effect on polypoid cells and adjacent normal colonic cells as shown in Figure 1 (*p* < 0.03). CBD-rich extract had a similar effect as synthetic CBD.

### 2.3. Evaluation of the Effect of Combinations of Cannabinoids on Colon Cancer Cell Viability

Various isolated cannabinoids were evaluated for synergistic interaction in inhibiting colon-polyp-derived cell viability. As shown in Table 3 and in Figure 2, Figure 3, Figure 4 and Figure 5, combinations between CBCA (at X2 concentration—29 µM or X1 concentration—14.5 µM), CBDV (at X2 concentration—47 µM or X1 concentration—23.5 µM), THCV (at X2 concentration—40/48 µM or X1 concentration—20/24 µM), and CBGA (at X2 concentration—51.2 µM or X1 concentration—25.6 µM), significantly inhibited the viability of the polyp-derived cells. The cannabinoid combinations CBCA + CBDV (*p* < 0.0001, Bliss equation calculation = 0.6), CBCA + THCV (*p* < 0.01, Bliss equation calculation = 0.48), CBDV + CBGA (*p* < 0.0004, Bliss equation calculation = 0.5), and THCV + CBGA (*p* < 0.0025, Bliss equation calculation = 0.45) presented a synergistic effect. However, the cannabinoid combinations THCV+CBDVA and THCV+ CBDV had no synergistic effect, as presented in Figure 6 and Figure 7 (*p* > 5, Bliss equation calculation = 0.01 and *p* > 5, Bliss equation calculation = −0.15, respectively). Notably, the cannabinoids CBCA and CBGA exhibited weak toxicity when administered as a single active compound (Table 1) and became significantly more toxic when combined with other cannabinoids. The nature of the interaction between isolated cannabinoids varied significantly when combined with different cannabinoids. For example, THCV presented a synergistic interaction when combined with CBGA (Figure 5), an additive interaction when combined with CBDVA (Figure 6), and an antagonist interaction when combined with CBDV (Figure 7).

## 3. Discussion

Previous studies have suggested that cannabinoids have cytotoxic activity in CRC [18,19,20,21]. Colorectal polyps are the primary premalignant precursors of CRC [4,5]. Therefore, to examine the possibility of the therapeutic or preventive potential of both cannabinoid extracts and isolated cannabinoids, their effect on biopsies of colorectal polyps and healthy tissue from patients scheduled for colonic polypectomy were analyzed. Biopsies of polyps and normal colon tissue of the same patient were exposed to a variety of cannabinoids in a range of concentrations.

The present study provides evidence of a cytotoxic activity of botanic cannabinoid extracts and synthetic cannabinoids on colorectal polyps. We were able to demonstrate that isolated cannabinoids presented different effects on the viability of cells derived from colorectal polyps. Some of the cannabinoids (mainly THC-d8 and THC-d9) were the most toxic and exhibited persistent toxicity in all polyps tested. Additionally, several of the cannabinoids exhibited a broad variability in their capacity to elicit toxicity when different polyps were tested.

In view of the differential effect of the various cannabinoids on different polyps, it may be necessary to initially test the effect of the extracts on polyps, prior to using them as therapeutic agents. Cells of a resected polyp could potentially be incubated with various cannabinoids to identify and select the agents that most effectively inhibit the viability of the lesion’s cells. Those toxic agents could be then prescribed for subsequent treatments by applying a pharmaceutical composition composed of one or more of the effective agents. Moreover, such personalized combinations, specifically those with differential activity against polyp-derived cells, could be administered for preventive purposes.

CBD was shown to have a differential cytotoxic effect in the presented study. It appeared to be more toxic to polypoid cells than to adjacent colonic cells at a concentration of 15 µM. Since nearly every carcinoma begins with a polyp [24], CBD could be a potential candidate as a chemopreventive agent to either prevent or suppress the progression of neoplastic polyps. The cytotoxic effect of CBD further suggests its therapeutic anticancer value. The differential effect of CBD could potentially allow local administration into the mucosa and/or submucosa layers during polypectomy to reduce the risk of recurrence without harming the adjacent normal colonic mucosa.

Combinations of the cannabinoids CBDV, THCV, CBDVA, CBCA, and CBGA exhibited a synergistic inhibitory effect on the viability of cells derived from colon polyps of patients. Previous studies have demonstrated a synergistic interaction of cytotoxic activity against colorectal polyps for several combinations [23]. Synergism between plant-produced compounds was previously suggested because in some cases, unrefined content of the flower extract with its different extracted compounds may have an advantage over the activity of an isolated compound [25,26,27]. It is well known that cancerous cells may develop resistance to anti-cancer drugs. Thus, utilizing combinations of cannabinoids may be of particular interest in the treatment of cancer in general, as well as in the inhibition of pre-cancerous lesions and avoidance of recurrence following endoscopic removal. Nevertheless, in vivo experiments of synergism would be necessary.

To the best of our knowledge, in vivo studies assessing the effect of local administration of cannabinoids to polyp resection beds have not been published to date. Two theoretical measures of safety, the use of non-psychotropic purified cannabinoids and synergism, should be provided for optimized general and local safety profiles. Moreover, the concentrations of cannabinoids used in the cell experiments (i.e., 50 µM) were orders of magnitude lower than those used in clinical trials (i.e., up to 3 mM/day), as shown in a recent publication [28]. Certainly, studies with both systemic and local administration of escalating doses of purified cannabinoid combinations will have to be conducted before moving forward to efficacy studies and human trials.

To conclude, our study results support the potential cytotoxic effect of cannabinoid extracts on colorectal polyps, as well as their synergistic and differential interactions. Further studies examining this postulation and the ultimate combination of cannabinoids for inhibiting/decreasing the recurrence rate of neoplastic polyps, and for preventing their malignant transformation into adenocarcinoma, are needed. Furthermore, the combined activity of selected cannabinoids with conventional chemotherapeutic agents, as well as biological selective agents, should be explored.

## 4. Materials and Methods

### 4.1. Polyp and Normal Colon Derived Cells

Cells were isolated from polyp specimens and from normal colon specimens. Biopsies from polyps and healthy colonic tissue from the same patient were obtained from twenty-two patients scheduled for polyp resection. Only patients with a large polyp that appeared benign endoscopically, was a granular lateral spreading tumor with 0-Is or 0-IIa morphology as per the Paris endoscopic classification [29], and was over 20 mm in size, which was found suitable for piecemeal endoscopic mucosal resection (EMR) by a gastroenterologist, were included. Exclusion criteria included: positive infectious serology (HAV, HBV, HCV, and HIV serologies were tested), polyp of less than 20 mm in size, malignant appearing polyp, polyp amendable for en bloc resection, a granular lateral spreading tumor with 0-IIb/0-IIc/0-III morphology as per the Paris endoscopic classification, a non-granular lateral spreading tumor pregnancy, breast-feeding, alcohol/drug abuse, minority, and impaired judgment. Overall, following final exclusion, twenty polyps from twenty patients were evaluated.

The study was conducted according to the guidelines of the Declaration of Helsinki and was approved by the Institutional Ethics Committee of Tel Aviv medical center (Helsinki approval no.TLV-0194-20). Informed Consent Statement: Informed consent was obtained from all subjects involved in the study.

Biopsies taken during each colonoscopy were placed in tissue culture media preservation media (MACS, Tissue storage solution, cat.130-100-008, Miltenyi Biotec, Bergisch Gladbach, Germany) and immediately transported to the laboratory. Dissociation of the biopsies’ tissue into single-cell suspensions for subsequent use was performed by using the MACS Technology by Miltenyi Biotec (Human Tumor Dissociation Kit cat. 130-095-929 and gentle MACS octo Dissociator) according to manufacturer’s instructions. After dissociation, cell viability was determined by using cell-permeable dye (Propidium Iodide, PI) and analyzed by flow cytometric analysis.

### 4.2. Polyp Derived Cells Viability and Counting

1 μL PI was diluted in 1 mL of 0.5%BSA/PBS buffer. A total of 90 μL diluted PI were added to 10 μL cells and was mixed well. Cell viability was analyzed by flow cytometric analysis (Attune NxT Flow Cytometer, Thermo Fisher Scientific, Waltham, MA, USA).

### 4.3. Extraction of Cannabis Inflorescence

Dry inflorescences of cannabis plants (CBD rich extract, RCK Ltd., Hof Ashkelon, Israel) were used in this experiment for the extraction of cannabis tinctures. The inflorescences were pulverized with liquid nitrogen and then extracted by press at 75 °C for 120 s. The decarboxylation of extracts was performed by heating for 2 h at 140 °C. The heated extracts contained mainly CBD (76%). The tincture was resuspended in dimethyl sulfoxide (DMSO) for a stock solution of 100 mM (control DMSO). For the treatments, the stock solution was diluted accordingly for cell cultures and biopsies in all experiments. Data regarding types of the CBD-rich extract as well as the percentage of cannabinoids as measured by HPLC chromatogram are presented in Table 4.

### 4.4. Isolated Cannabinoids

Isolated cannabinoids were purchased from RESTEK for analytical purposes as HPLC markers and for research. The cannabinoid purity was > 99%. The IC_50_, as measured in our laboratory on many cell lines, ranged from 15–100 µM. Based on this measurement, the starting concentrations for polyp experiments were determined. Later, these concentrations were adjusted to obtain the concentrations defined as 1X and 2X for each cannabinoid. Due to the very small number of cells obtained from one polyp, the ability to perform experiments with different concentrations was limited, as well as the variation from polyp to polyp. Therefore, an IC_50_ test was not performed for these cannabinoids on polyps; hence, we performed toxicity experiments at uniform cannabinoid concentrations and synergy experiments. The cannabinoids were dissolved in DMSO for a stock solution of 20 mM. The tested cannabinoid concentration did not exceed 60 µM and the DMSO concentration did not exceed 0.3%. This concentration of DMSO showed no toxic effect on the cells in these experiments. The cannabinoids and extracts were analyzed by HPLC (SHIMADZU, Nexera-i—LC-2040C Plus).

### 4.5. Cytotoxicity Assay

CTG (CellTiter-Glo Luminescent Cell Viability Assay, G7573, Promega) was used to check the cytotoxic effect of the cannabinoids and extracts. CTG was added to each well of the treatments and incubated at room temperature for 10 min. The luminescence was then recorded. The percentage of live cells was calculated relative to the nontreated control wells.

### 4.6. Cannabinoids Extracts and Compounds Cytotoxicity

To evaluate cannabinoids’ toxicity effects, about 4000–5000 polyp-derived cells were plated into each well in a 384-well plate. Cells were incubated for 48 h at 37 °C with various purified cannabinoids or extracts at different concentrations. The experiment sets were calibrated extensively on many cell lines and different types of cancerous and adjacent normal tissue. Time from 12–72 h was evaluated. The time point of 48 h of incubation was defined as the preferred time in terms of assay sensitivity and allowed the cells to perform normal behavior, such as cell division attempts, protein synthesis, etc. Cell viability (toxicity) was performed by CTG assay.

### 4.7. Cannabinoids Differential Effects

Polyp and adjacent normal colonic tissues were obtained following polyp resection. The differential toxic effect of the extracts on polyp cells as compared to normal colonic tissue cells was evaluated using an assessment of overall cell viability by CTG assay.

### 4.8. Cannabinoids Combined Effects (Synergy)

To determine whether the interaction of different cannabinoids was synergistic, that is, their combined activity was greater than the sum of their separate activities, the extent of activity in different combined concentrations of cannabinoids was examined. To examine synergy in cytotoxic activity between cannabinoids, a CTG assay was used on biopsy cells as described above. For the synergy experiments, cannabinoids were used at a concentration of 1× (viability above 85%) and 2×. For the synergic assay, we used 1× of A, 1× of B, and 1× of A + B. The synergistic effect was calculated according to cells’ survival experimental results according to the Bliss Equations (1) and (2) as follows:(1)Survival=S,SAB(Cal)=SA×SB
(2)Synergy(>0)=SAB(Cal)−SAB(Exp)

S designates survival, wherein SAB (calculated) is a calculation of survival, e.g., of cells exposed to a combination of cannabinoid A and cannabinoid B, SA is the survival of cells exposed to cannabinoid A only, and SB is the survival of cells exposed to cannabinoid B only. SAB (Exp) designates the observed survival of cells exposed to the combination of cannabinoid A and cannabinoid B.

### 4.9. Statistical Analyses

Student’s *t*-test was used for *p* value assessment. Statistical analysis was performed on the results by performing Student’s *t*-Test (Excel Analysis ToolPak).

## 5. Patents

International Patent Application, publication No.: WO/2011/110866 discloses the use of phytocannabinoids, either in an isolated form or in the form of a botanical drug substance (BDS) in the treatment of cancer (https://patents.google.com/patent/WO2021245677A1/en, accessed on 5 June 2020).

## Figures and Tables

**Figure 1 ijms-23-11366-f001:**
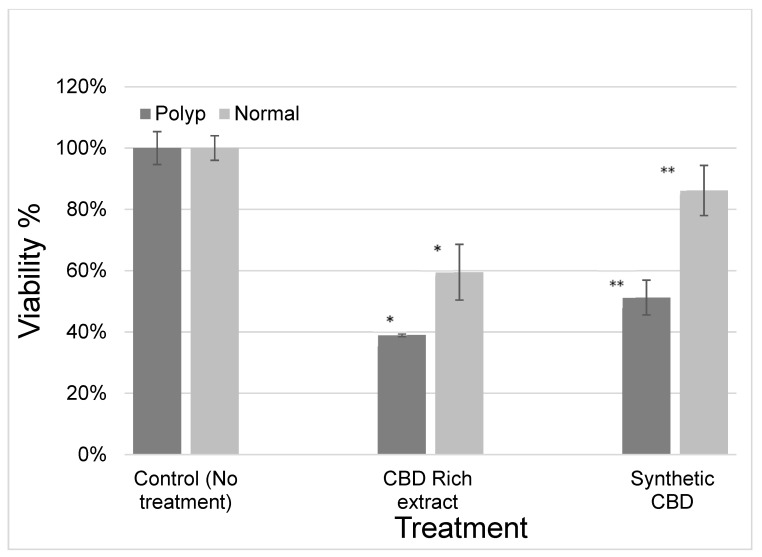
Influence of cannabidiol (CBD) and CBD-rich extract on the viability of polypoid cells and adjacent normal colonic cells. CBD-rich extract had a similar effect as synthetic CBD. * *p* < 0.03, ** *p* < 0.02.

**Figure 2 ijms-23-11366-f002:**
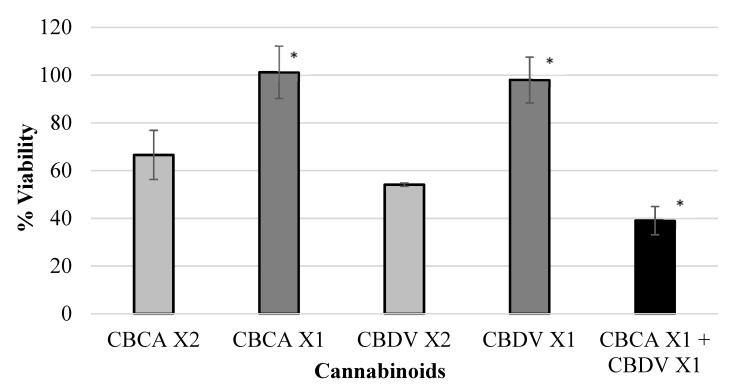
Viability of polyp cells exposed to CBDV and CBCA. Combinations between CBCA (at X2 concentration—29 µM or X1 concentration—14.5 µM) and CBDV (at X2 concentration—47 µM or X1 concentration—23.5 µM) significantly inhibited the viability of polyp-derived cells. The cannabinoid combination CBCA + CBDV presented a synergistic effect (* *p* < 0.0001, Bliss equation calculation = 0.6). Legend: cannabichromene acid (CBCA), cannabidivarin (CBDV).

**Figure 3 ijms-23-11366-f003:**
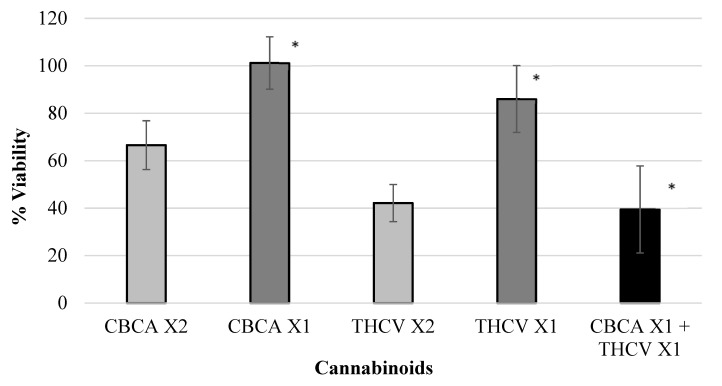
Viability of polyp cells exposed to CBCA and THCV. Combinations between CBCA (at X2 concentration—29 µM or X1 concentration—14.5 µM) and THCV (at X2 concentration—40/48 µM or X1 concentration—20/24 µM) significantly inhibited the viability of polyp-derived cells. The cannabinoid combination CBCA + THCV presented a synergistic effect (* *p* < 0.01, Bliss equation calculation = 0.48). Legend: cannabichromene acid (CBCA), tetrahydrocannabivarin (THCV).

**Figure 4 ijms-23-11366-f004:**
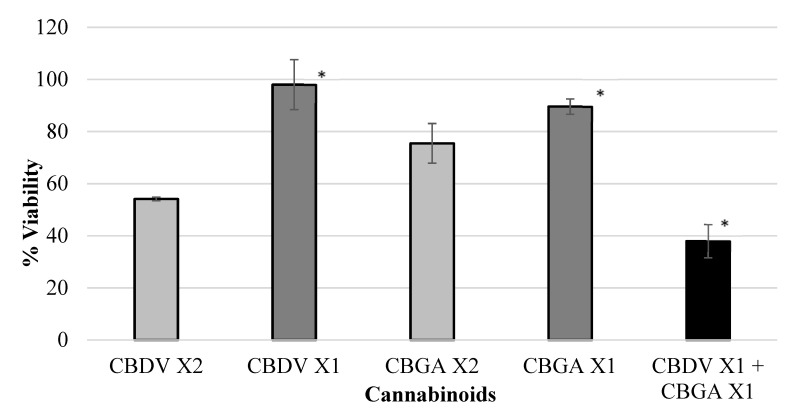
Viability of polyp cells exposed to CBDV and CBGA. Combinations between CBDV (at X2 concentration—47 µM or X1 concentration—23.5 µM) and CBGA (at X2 concentration—51.2 µM or X1 concentration—25.6 µM) significantly inhibited the viability of polyp-derived cells. The cannabinoid combination CBDV + CBGA presented a synergistic effect (* *p* < 0.0004, Bliss equation calculation = 0.5). Legend: cannabidivarin (CBDV), cannabigerolic acid (CBGA).

**Figure 5 ijms-23-11366-f005:**
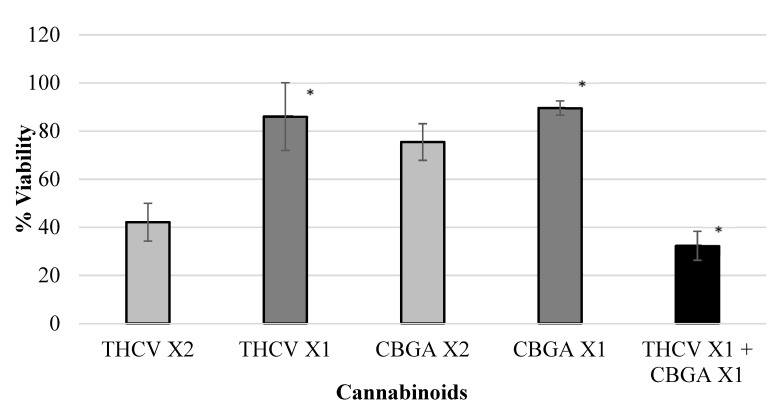
Viability of polyp cells exposed to THCV and CBGA. Combinations between THCV (at X2 concentration—40/48 µM or X1 concentration—20/24 µM) and CBGA (at X2 concentration—51.2 µM or X1 concentration—25.6 µM) significantly inhibited the viability of polyp-derived cells. The cannabinoid combination THCV + CBGA (* *p* < 0.0025, Bliss equation calculation = 0.45) presented a synergistic effect. Legend: tetrahydrocannabivarin (THCV), cannabigerolic acid (CBGA).

**Figure 6 ijms-23-11366-f006:**
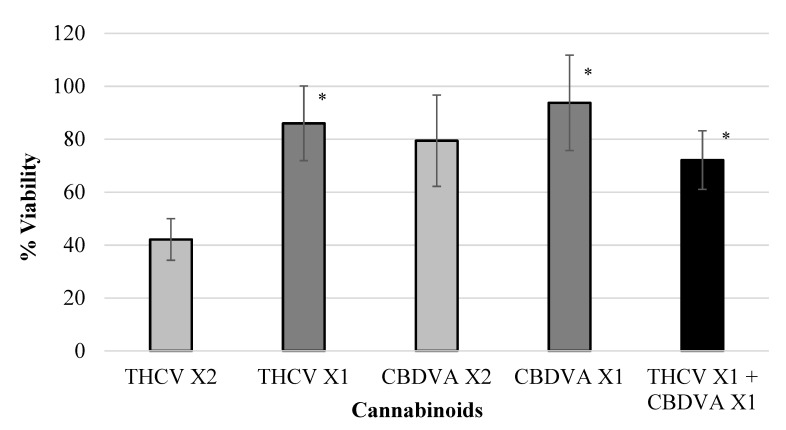
Viability of polyp cells exposed to THCV and CBDVA. The cannabinoid combination THCV + CBDVA showed no synergistic effect (* *p* > 5, Bliss equation calculation = 0.01). Legend: tetrahydrocannabivarin (THCV), cannabidivarinic acid (CBDVA).

**Figure 7 ijms-23-11366-f007:**
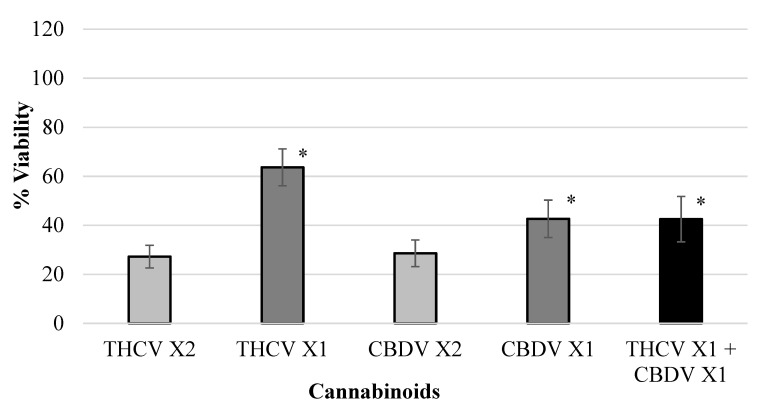
Viability of polyp cells exposed to THCV and CBDV. The cannabinoid combination THCV + CBDV had no synergistic effect (* *p* > 5, Bliss equation calculation = −0.15). Legend: tetrahydrocannabivarin (THCV), cannabidivarin (CBDV).

**Table 1 ijms-23-11366-t001:** Patient and polyp characteristics.

Patient No.	Sex (F = Female, M = Male)	Age	Histology	Polyp Site	Polyp Size
1	F	57	VA w LGD	Cecum	25 mm
2	M	69	TVA w LGD	Cecum	20 mm
3	M	77	TVA w HGD	Descending colon	40 mm
4	F	52	TSA	Sigmoid colon	70 mm
5	F	50	VA w LGD	Sigmoid colon	50 mm
6	M	62	TVA w LGD	Ascending colon	30 mm
7	F	66	SSA w LGD	Transverse colon	25 mm
8	F	64	TVA w LGD	Ascending colon	30 mm
9	M	60	TVA w LGD	Ascending colon	20 mm
10	M	62	TVA w LGD	Ascending colon	40 mm
11 *	M	80	TVA w LGD	Cecum	20 mm
12	F	55	TVA w HGD	Ascending colon	30 mm
13	F	50	SSA no dysplasia	Ascending colon	20 mm
14	F	74	SSA no dysplasia	Ascending colon	20 mm
15 *	F	73	VA w LGD	Ascending colon	40 mm
16	M	63	TA w HGD	Cecum	50 mm
17	F	71	TA w LGD + SSA	Splenic flexure	25 mm
18	M	60	TA w LGD	Ascending colon	25 mm
19	M	80	VA w LGD	Hepatic flexure	30 mm
20	F	66	VA w LGD	Cecum	70 mm
21	M	51	TVA w LGD	Rectum	30 mm
22	M	80	TA w LGD	Transverse colon	30 mm

* Patient polyp numbers 11 and 15 were excluded from study analysis due to borderline-positive hepatitis A virus (HAV) serology. Legend: tubular adenoma (TA), tubulovillous adenoma (TVA), villous adenoma (VA), traditional serrated adenoma (TSA), sessile serrated adenoma (SSA), low grade dysplasia (LGD), high grade dysplasia (HGD).

**Table 2 ijms-23-11366-t002:** Toxicity of cannabinoids on colon-polyp-derived cells.

Cannabinoid	P004	P005	P006	P008	P009	P010	Average Viability
THC-d8	10%	23%	44%	4%	7%	5%	15%
THC-d9	20%	20%	47%	13%	21%	8%	21%
CBC	21%	21%	61%	15%	22%	11%	25%
CBN	24%	32%	54%	18%	16%	13%	26%
CBDV	37%	63%	62%	23%	39%	9%	39%
CBD	21%	41%	54%	35%	62%	24%	40%
THCV	59%	74%	70%	47%	40%	18%	51%
CBL	73%	55%	89%	55%	63%	29%	61%
CBG	75%	69%	58%	69%	65%	37%	62%
THCA	59%	91%	99%	52%	60%	32%	66%
CBDA	104%	97%	66%	38%	73%	26%	67%
CBDVA	72%	90%	83%	48%	69%	56%	70%
CBNA	66%	94%	76%	83%	70%	60%	75%
CBCA	63%	86%	105%	91%	72%	80%	83%
CBGA	71%	96%	84%	116%	70%	97%	89%

Toxicity of cannabinoids on colon-polyp-derived cells (P004–P010 designate the various patients’ polyps), shown as percentage of survival in increasing order. Average survival is shown on the rightmost column. Legend: tetrahydrocannabinol (THC), cannabichromene (CBC), cannabinol (CBN), cannabidivarin (CBDV), cannabidiol (CBD), tetrahydrocannabivarin (THCV), cannabicyclol (CBL), cannabigerol (CBG), tetrahydrocannabinolic acid (THCA), cannabidiolic acid (CBDA), cannabidivarinic acid (CBDVA), cannabinolic acid (CBNA), cannabichromene acid (CBCA), cannabigerolic acid (CBGA).

**Table 3 ijms-23-11366-t003:** The effect of combinations of cannabinoids on polyp cells’ viability.

Cannab. A	Cannab. B	Cells Viability at ×2 Conc. of Cannab. A	Cells Viability at ×1 Conc. of Cannab. A	Cells Viability at ×2 Conc. of Cannab. B	Cells Viability at ×1 Conc. of Cannab. B	Cells Viability at ×1 Conc. of Cannab. A + Cannab. B
CBCA	CBDV	67	101	54	98	39
CBDV	CBDVA	54	98	79	94	39
CBDV	CBGA	54	98	75	90	38
CBCA	THCV	67	101	42	86	39
CBCA	CBGA	67	101	75	90	61
CBGA	THCV	75	90	42	86	32
CBCA	CBDVA	67	101	79	94	79
CBDVA	CBGA	79	94	75	90	61
CBDVA	THCV	79	94	42	86	72

The cannabinoid combinations: CBCA + CBDV, CBCA + THCV, CBDV + CBGA, and CBGA + THCV presented synergistic effects. Legend: cannabichromene acid (CBCA), cannabidivarin (CBDV), cannabigerolic acid (CBGA), cannabid varinic acid (CBDVA).

**Table 4 ijms-23-11366-t004:** Cannabinoid percentage in CBD-rich extract tincture.

CBDV	THCV	CBD	CBG	CBDA	CBGA	CBN	THC-d9	THC-d8	CBC	THCA	CBDVA	CBCA
0.3	3.1	76.0	1.6	0.1	0.1	0.1	2.9	0.0	5.6	0.0	0.0	5.9

Types and percentage of cannabinoids as measured by HPLC chromatogram. Legend: cannabidivarin (CBDV), tetrahydrocannabivarin (THCV), cannabidiol (CBD), cannabigerol (CBG), cannabidiolic acid (CBDA), cannabigerolic acid (CBGA), cannabinol (CBN), tetrahydrocannabinol (THC), cannabichromene (CBC), tetrahydrocannabinolic acid (THCA), cannabidivarinic acid (CBDVA), cannabichromene acid (CBCA).

## Data Availability

Not applicable.

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
