# Peer review of "The Cytotoxic Effect of Isolated Cannabinoid Extracts on Polypoid Colorectal Tissue"

_ijms, 2022, doi:10.3390/ijms231911366_

Round 1

Reviewer 1 Report

The Authors report a study on the effects of several cannabinoids on the viability of cells derived from colorectal polyps and normal mucosa.

The study is in line with recent research on the effects of cannabinoids on different metabolic pathways in humans. In this case a possible anticancer effect is tested. The results are rather interesting though the presentation of them is not optimal.

In particular the questions are:

1       It seems that the cannabinoids THC-d8 and THC-d9 are the most effective in reducing viability of cells (Table 1). However their efficacy in combination with other molecules is not reported. The Authors should explain why.

2.     On what ground have Authors chosen the concentrations of cannabinoids to be tested (i.e., previous works, previous experiments on cytotoxicity…) ?

3.       In table 3, column 4, a viability of more than 100% is reported.at concentration X1 for 4 cannabinoid A. What does it mean ?

4.     Figure 2. For all the 4 panels the p values are not reported on the bars of the graphic, thus it is impossible to understand differences. It is necessary to review the figure.

5.      Most of the conclusions of the discussion are based on results obtained on a limited number of polyps (see table 1). Indeed, in the methods section it is not clear how many polyps were used to derive cells for the experiments. I think that the section should be changed in order to include these informations.

6.       In the discussion section some other points should be discussed, i.e., are there some data to predict the possible general toxicity to the patient when in vivo studies will be performed? How long the effect of cannabinoid extracts could last?

 Some English polishing is needed (i.e., saxon genitives and minor errors)

Author Response

Dear reviewer,

Thank you for the highly valued and important comments.

The requested changes were made in the revised manuscript.

Comments from Reviewer 1:

  1. Comment 1: It seems that the cannabinoids THC-d8 and THC-d9 are the most effective in reducing viability of cells (Table 1). However their efficacy in combination with other molecules is not reported. The Authors should explain why.

Response: Thank you for pointing this out. Synergy and activity of THC-d8 and THC-d9 were tested on many different cell lines for research purposes. Since this research is aimed at the practical purpose of using these substances in patients, we decided to exclude psychoactive substances or those that could lead to an immediate regulatory difficulty.

  1. Comment 2:On what ground have Authors chosen the concentrations of cannabinoids to be tested (i.e., previous works, previous experiments on cytotoxicity…)?

Response: The concentrations of cannabinoids that were tested were chosen based on previous cytotoxicity experiments on nearly 30 cell lines from different tissue sources (not in the scope of this article).

  1. Comment 3:In table 3, column 4, a viability of more than 100% is reported.at concentration X1 for 4 cannabinoid A. What does it mean ?

Response: The maximum percentage of viability is determined by normalization of the treated samples to control samples of cells that did not receive treatment. This is 100% viability. Under standard biological experimental conditions on cell lines and certainly on cells derived from biopsies, there is a non-significant variation arising from the experimental conditions and not from the tested material.

  1. Comment 4:Figure 2. For all the 4 panels the p values are not reported on the bars of the graphic, thus it is impossible to understand differences. It is necessary to review the figure.

Response: Thank you for highlighting this point. All 4 panels of Figure 2 were corrected in the revised manuscript.

  1. Comment 5:Most of the conclusions of the discussion are based on results obtained on a limited number of polyps (see table 1). Indeed, in the methods section it is not clear how many polyps were used to derive cells for the experiments. I think that the section should be changed in order to include these informations.

Response: Thank you for your comment. We revised the following paragraph in the method section- “Cells were isolated from polyp specimens and from normal colon specimens. Biopsies from polyps and healthy colonic tissue from the same patient were obtained from twenty-two patients scheduled for polyp resection. Only patients with a large polyp that appears benign endoscopically, a granular lateral spreading tumor with 0-Is or 0-IIa morphology as per the Paris endoscopic classification [24], over 20 mm in size, which was found suitable for piecemeal endoscopic mucosal resection (EMR) by a gastroenterologist were included. Exclusion criteria included: positive infectious serology (HAV, HBV, HCV, and HIV serologies were tested), polyp of less than 20 mm in size, malignant appearing polyp, polyp amendable for en bloc resection, a granular lateral spreading tumor with 0-IIb/0-IIc/0-III morphology as per the Paris endoscopic classification, a non-granular lateral spreading tumor, pregnancy, breast-feeding, alcohol/drug abuse, minority, and impaired judgment. Overall, following final exclusion, twenty polyps from twenty patients were evaluated.”

  1. Comment 6:In the discussion section some other points should be discussed, i.e., are there some data to predict the possible general toxicity to the patient when in vivo studies will be performed? How long the effect of cannabinoid extracts could last?

Response: To the best of our knowledge, in vivo studies assessing the effect of local administration of cannabinoids to polyp resection beds have not been published to date. Two theoretical measures of safety, the use of non-psychotropic purified cannabinoids and synergism should provide for an optimized general and local safety profiles. Moreover, the concentrations of cannabinoids used in the cell experiments (i.e. 50 µM) were orders of magnitude lower than those used in clinical trial (i.e. up to 3mM/day), as shown in a recent publication. (Dosage, Efficacy and Safety of Cannabidiol Administration in Adults: A Systematic Review of Human Trials. J Clin Med Res. 2020;12(3):129-141). Certainly, studies with both systemic and local administration of escalating doses of purified cannabinoid combinations will have to be conducted before moving forward to efficacy studies and human trials.

** Some English polishing is needed (i.e., saxon genitives and minor errors) –

Additional editing was done, corrections were made in the revised version.

Reviewer 2 Report

The Authors presented an interesting paper regarding potential therapeutic efficacy of cannabinoids on colorectal cancer cells. However, before this paper is published, some issues need to be corrected and /or precisely described. Otherwise, this manuscript is quite chaotic.  

1. The first sentence in the introduction section should be first started with the full "name" of the tumor, not the abbreviation (CRC)

2. Please provide whether cannabinoids extracts that were determined for their cytotoxic effect in vitro were used as a DMSO solution. Unfortunately, DMSO is highly harmful for cells even in a low concentration. Therefore, please state in the methodology what was the final concentration of DMSO as a solution for cannabinoid extracts. 

3. The Authors performed the cytotoxic analysis within 48 h. Do the Authors  have any results for 24 or 72 h? If not, please provide the reason for which the Authors did not try to observe the duration of cannabinoids efficacy?

4. Although the Authors used normal colonic tissues as a control group, in my opinion it would be nice (and for sure the paper would be much improved) if the Authors could provide with the results for a well-known drug with anticancer effect towards CRC (as a comparison and control).

5. The Authors used various concentration of cannabinoids and/or the extracts. In line with this, please provide the IC50 for each drug/extract.

6. Unfortunately, I did not find any approval of the Bioethics committee which should be given when studies on humans and human-sourced items are stated.

7. Since the results demonstrate patients, their characteristic, inclusion and exclusion criteria should be provided in the methodology section.

8. Table one presents single cannabinoids, though the Authors written that also extracts were evaluated. Therefore, please provide types of the extracts, eventual chromatograms with their characteristic (e.g., CBD-rich extract, etc.)

9. Since various cannabinoids and their toxicity are given in the Table 1, the Authors should indicate these cannabinoids biological behavior (agonist/antagonist/partial agonist) towards CB receptors. This would clearly present what type of CB receptors together with the type of the ligand is crucial for the induction of anticancer effect in CRC.

10. What was the reason to four on CBD and CBD-rich extract while presenting cell viability. This should be provided.

11. Also, when showing statistics, please provide the type of the test and post-test.

12. I'm confused with the results demonstrating the character of interactions between the cannabinoids. Therefore, please describe clearly on what basis did the Authors indicate synergy/additive action or inhibition?

13. Please define PI.

Author Response

Dear reviewer,

Thank you for the highly valued and important comments.

The requested changes were made in the revised manuscript.

  1. Comment 1:The first sentence in the introduction section should be first started with the full "name" of the tumor, not the abbreviation (CRC).

Response: Thank you for the comment. The requested change was made in the introduction section.

  1. Comment 2:Please provide whether cannabinoids extracts that were determined for their cytotoxic effect in vitro were used as a DMSO solution. Unfortunately, DMSO is highly harmful for cells even in a low concentration. Therefore, please state in the methodology what was the final concentration of DMSO as a solution for cannabinoid extracts. 

Response: In these experiments, the tested cannabinoid concentration did not exceed 60 µM and the DMSO concentration did not exceed 0.3%. This concentration of DMSO shows no toxic effect on the cells in these experiments. Data was added to the revised methods section.

  1. Comment 3:The Authors performed the cytotoxic analysis within 48 h. Do the Authors have any results for 24 or 72 h? If not, please provide the reason for which the Authors did not try to observe the duration of cannabinoids efficacy?

Response: The experiment sets were calibrated extensively on many cell lines and different types of cancerous and adjacent normal tissue, in our lab. Time from 12-72h was evaluated. The time point of 48h incubation, was defined as the preferred time in terms of assay sensitivity and allows the cell to perform normal behavior, such as cell division attempts, protein synthesis, and more.  In addition, in unpublished experiments in our lab of time duration and mode of action analysis (not in the scope of this article), there were very few cannabinoids which worked in the first 24h but for all of them the influence can see in 48h. 72h shows no added benefit to the assay interpretation. A comment was added to the revised methods section.

  1. Comment 4:Although the Authors used normal colonic tissues as a control group, in my opinion it would be nice (and for sure the paper would be much improved) if the Authors could provide with the results for a well-known drug with anticancer effect towards CRC (as a comparison and control).

Response: Thank you for the significant comment. Currently we aim to explore the synergistic effect of cannabinoids and several commonly used chemotherapeutic agents on polyps’ viability. However, unfortunately we can’t yet provide the results at this preliminary stage.

  1. Comment 5:The Authors used various concentration of cannabinoids and/or the extracts. In line with this, please provide the IC50 for each drug/extract.

Response: The IC50 as measured in our laboratory on many cell lines ranges from 15-100 µM. Based on this measurement, the starting concentrations for polyp experiments were determined. Later, these concentrations were adjusted to obtain the concentrations defined as 1X and 2X for each cannabinoid. Due to the very small amount of cells obtained from one polyp, the ability to perform experiments with different concentrations is limited, as well as the variation from polyp to polyp. Therefore, an IC50 test was not performed for these cannabinoids on polyps; hence, we performed toxicity experiments at uniform cannabinoid concentrations and synergy experiments. A comment was added to the revised methods section.

  1. Comment 6:Unfortunately, I did not find any approval of the Bioethics committee which should be given when studies on humans and human-sourced items are stated.

Response: 2nd paragraph in the revised methods section: The study was conducted according to the guidelines of the Declaration of Helsinki, and approved by the Institutional Ethics Committee of Tel Aviv medical center (Helsinki approval no. 0194-20-TLV). Informed consent was obtained from all subjects involved in the study. Helsinki approval no. is provided in the methods section.

  1. Comment 7:Since the results demonstrate patients, their characteristic, inclusion and exclusion criteria should be provided in the methodology section.

Response: Thank you for your comment. We revised the following paragraph in the method section- “Cells were isolated from polyp specimens and from normal colon specimens. Biopsies from polyps and healthy colonic tissue from the same patient were obtained from twenty-two patients scheduled for polyp resection. Only patients with a large polyp that appears benign endoscopically, a granular lateral spreading tumor with 0-Is or 0-IIa morphology as per the Paris endoscopic classification [24], over 20 mm in size, which was found suitable for piecemeal endoscopic mucosal resection (EMR) by a gastroenterologist were included. Exclusion criteria included: positive infectious serology (HAV, HBV, HCV, and HIV serologies were testes), polyp of less than 20 mm in size, malignant appearing polyp, polyp amendable for en bloc resection, a granular lateral spreading tumor with 0-IIb/0-IIc/0-III morphology as per the Paris endoscopic classification, a non-granular lateral spreading tumor, pregnancy, breast-feeding, alcohol/drug abuse, minority, and impaired judgment. Overall, following final exclusion, twenty polyps from twenty patients were evaluated.” Data regarding patients and polyps’ characteristic is provided in Table 1.

  1. Comment 8:Table one presents single cannabinoids, though the Authors written that also extracts were evaluated. Therefore, please provide types of the extracts, eventual chromatograms with their characteristic (e.g., CBD-rich extract, etc.)

Response: Data regarding types of the extract as well as the percentage of cannabinoids as measured by HPLC chromatogram was added in the method section (table 4).

Cannabinoid percentage in CBD-rich extract Tincture

CBDV

THCV

CBD

CBG

CBDA

CBGA

CBN

THC-d9

THC-d8

CBC

THCA

CBDVA

CBCA

0.3

3.1

76.0

1.6

0.1

0.1

0.1

2.9

0.0

5.6

0.0

0.0

5.9

  1. Comment 9:Since various cannabinoids and their toxicity are given in the Table 1, the Authors should indicate these cannabinoids biological behavior (agonist/antagonist/partial agonist) towards CB receptors. This would clearly present what type of CB receptors together with the type of the ligand is crucial for the induction of anticancer effect in CRC.

Response: We appreciate the significance of attaching mechanistic explanations to empiric findings, and indeed there are numerous publications discussing the interactions of various cannabinoids with different cell lines/tissues. However, since we chose to test cannabinoid combinations empirically based on previously conducted cell line experiment, we thought that diving into mechanistic interaction of specific cannabinoids with the CB receptors would be overreaching at this point. Experiments conducted to elucidate mechanisms of action will be pursued after we zone in on our preferred cannabinoid combination for in vivo trials.

  1. Comment 10:What was the reason to four on CBD and CBD-rich extract while presenting cell viability. This should be provided.

Response: At similar concentrations, the toxicity of CBD-rich extract is usually greater than that of pure CBD, and this is stated in many articles probably due to the entourage effect, i.e. the effect of additional substances in the extracts. Here, we wanted to show that the differential effect of CBD that damages the polyp cells more than the cells from the normal tissue also appears in the extracts. Perhaps in the future, we will see the use of pure substances for treatment or even directly with certain extracts from the cannabis plant.

  1. Comment 11:Also, when showing statistics, please provide the type of the test and post-test.

Response: t-test was used for p value assessment. Statistical analysis was performed on the results by performing Student's t-Test (Excel Analysis ToolPak). Changes were made in the methods (Statistical analyses) section.

  1. Comment 12:I'm confused with the results demonstrating the character of interactions between the cannabinoids. Therefore, please describe clearly on what basis did the Authors indicate synergy/additive action or inhibition?

Response: The cannabinoids do not have the same toxic effect; therefore, for synergy

experiments, a different concentration for each cannabinoid was chosen. A

concentration that gives more than 85% viability was defined as concentration 1X. As

mentioned in the methods section, for the synergy experiments, we use 1X of A, 1X

of B, and 1X A + 1X B. The synergistic effect was calculated according to cells

survival experimental results according to the Bliss equation as follows:

Interaction effect = SAB (Cal) - SAB (Exp)

Survival= S,

SAB (Cal) = SA × SB; where SAB (Cal) is the calculated additive survival effect of the

drugs A and B as predicted by their individual experimental effects (SA and SB). SA is

the survival of cells exposed to cannabinoid A only; SB is the survival of cells

exposed to cannabinoid B only.

SAB (Exp) is the experimental observed additive survival of cells exposed to the

combination of cannabinoid A and cannabinoid B together.

In case the observed value is less than the calculated one, the combination

treatment is considered synergistic. In case the observed value is greater than the

calculated value, the combination treatment is considered antagonistic. If both values

are equal, the combination treatment is considered additive. Synergy (Interaction

effect >0); Additive (Interaction effect = 0); Antagonist (Interaction effect < 0).

We focused on very strong synergistic combinations, i.e. combinations that gave a

result that was at least as strong as the 2X the concentration of each substance

separately.

  1. Comment 13:Please define PI.

Response: Propidium Iodide = PI. Data is provided in the methods section.

Round 2

Reviewer 1 Report

The manuscript has been revised according to the raised questions. For me, the paper is acceptable for publication.

Reviewer 2 Report

The paper was improved by the Authors, though some issues remain "untouched", such as IC50 etc. Nonetheless, in my opinion this manuscript is now ready to be published